# Drops of Capillary Blood Are Not Appropriate for Hemoglobin Measurement with Point-of-Care Devices: A Comparative Study Using Drop Capillary, Pooled Capillary, and Venous Blood Samples

**DOI:** 10.3390/nu14245346

**Published:** 2022-12-16

**Authors:** Vanessa De la Cruz-Góngora, Ignacio Méndez-Gómez-Humarán, Elsa Berenice Gaona-Pineda, Teresa Shamah-Levy, Omar Dary

**Affiliations:** 1Centre for Evaluation and Survey Research, National Institute of Public Health, Cuernavaca 62100, Mexico; 2Centre for Research in Mathematics CIMAT, Aguascalientes Unit, Aguascalientes 20200, Mexico; 3Bureau for Global Health, USAID, Washington, DC 20523, USA

**Keywords:** HemoCue, hemoglobin, capillary blood, venous blood, cyanmethemoglobin, anemia

## Abstract

Population-based surveys matched by time but using different methodologies for determining hemoglobin (Hb) concentration have shown inconsistencies in estimating anemia prevalence. This study aimed to estimate measurement errors in Hb quantification in HemoCue 201+ using venous blood (VB) and capillary blood both drops (DCB) and pools (PCB), and compare the results against those of a reference method (VB analyzed in hematology analyzers based on the cyanmethemoglobin method). Children (*n* = 49), adult females (*n* = 50), and older adults (*n* = 50) were randomly allocated to donate VB (4 mL) and either DCB (three drops) or PCB (350 µL). Results in HemoCue were analyzed through Bland Altman and Lyn’s concordance against Hb concentration by the reference method. A positive average bias (systematic error) was found for the HemoCue (0.31 g/dL) using the same VB samples. This value was then subtracted from all readings carried out in the device. After this adjustment, DCB still produced a positive bias (0.42 ± 0.81 g/dL), and the variation of single results was ±1.6 g/dL (95% CI). PCB and VB performed similarly; the average bias was negligible (−0.02 ± 0.36 and 0.00 ± 0.33 g/dL, respectively) and the variation of the results (95% CI) was ±0.7 g/dL or lower. Lyn’s concordance values were 0.86, 0.96, and 0.98 for DCB, PCB, and VB, respectively. Random variation using DCB is too large to approximate the true Hb values, and therefore DCB should be discontinued for diagnosing anemia both in individuals and in populations.

## 1. Introduction

Accurate and precise anemia diagnosis among individuals and among population strata is essential to the design, monitoring, and evaluation of interventions and public policies associated with the reduction of this global health problem [1,2].

Today, point-of-care hemoglobinometes (mostly HemoCue brand) are commonly used for quick and simple determinations of hemoglobin (Hb) concentration. The model 201+ is factory calibrated against the hemoglobincyanide (HiCN) method [3], and has a stated accuracy of ±1.5% [4], which means that the systematic average bias of this equipment is within ±0.10 to ±0.20 g/dL of values obtained in a reference method. The precision of the single results depends on the performance skills of the technicians, and it is not commonly estimated. 

HemoCues are used extensively in population-based surveys to determine anemia prevalence using one drop of capillary blood (either from the heel for children younger than one year of age, or through finger prick for older individuals). However, values for anemia prevalence using this methodology have generally been much higher than those using venous blood in the same populations and in similar dates, as documented in Nepal, Jordan, Guatemala, Malawi, India, and other countries [5,6,7,8,9]. Recently, Hruschka et al. [10] reported large inconsistencies in Hb distribution and anemia estimation (from 9 to 31 percentage point differences) using capillary versus venous blood in two types of national representative surveys: the Demographic Health Survey (DHS) and the Biomarkers Reflecting Inflammation and Nutritional Determinants of Anemia (BRINDA) project, as paired by country and time using HemoCue devices. Previously, other authors have discouraged the use of finger-prick blood to determine Hb concentration [11,12] because of the large differences of results against venous blood or in consecutive drops of blood collected from the same individuals. Other researchers have evaluated the performance of Hb estimation from capillary pooled blood [11,13], producing results that may have practical applications in population-based surveys when collection of venous blood is difficult. Nevertheless, finger-prick blood is still commonly used for Hb determination worldwide.

In Mexico, we found anemia trends with inconsistences that are very difficult to interpret. Moreover, Mexican surveys matched by time, ENIM 2015 and ENSANUT 2016, using the same methodology for Hb estimation (Hemocue 201+, capillary drop blood) show a different prevalence of anemia in children (14.1% and 26.9%, respectively) [14,15]. 

In this study, we aimed to estimate measurement errors in Hb determination using HemoCue 201+ in venous and capillary blood (both drops and pooled), and compared the results against venous blood analyzed in clinical hematology analyzers (i.e., hemocounters). In this study, our reference constituted Hb results estimated with venous blood using the cyanmethemoglobin method. Venous blood samples and either drops of capillary blood or pools of capillary blood were collected from the same individual and across groups including children 1–4 years, adult non-pregnant females (18–49 years), and older adults (over 60 years).

## 2. Materials and Methods

### 2.1. Sampling Desing

This cross-sectional study was carried out between November and December 2020 in Jojutla, which is a small town at 970 m above sea level in Morelos, Mexico. Samples from three different groups were obtained: 49 children (1–4 years), 50 non-pregnant adult females (18–49 years old), and 50 older adults (60 years and older). Every individual donated a sample of 4 mL of venous blood that were collected in vacutainer tubes containing K-EDTA as anticoagulant, half of them also provided single drops of capillary blood by finger prick, while for the other half several drops of finger-pricked blood were pooled by collecting and mixing them inside a microtainer with an anticoagulant. 

Exclusion criteria included the presence of fever, cold, suspected exposure to or symptoms of COVID-19; being a female who was pregnant, lactating, or with a history of cancer, chemotherapies, or mastectomy; or self-report of any disease related to hematological disorders. For all individuals, a willingness to participate declaration was obtained.

Participants with Hb measurement below 8 g/dL in the first HemoCue measurement using capillary blood (drops or pooled) were excluded and referred to a health care provider for further examination.

Blood samples were collected by six trained and standardized personnel with previous experience collecting blood samples. Only one HemoCue apparatus per each surveyor was used for every aspect of the study. Personnel were instructed to collect blood samples in only one attempt, either through the finger (capillary blood) or the left arm (venous blood). In the case of the group designed to collect pools of capillary blood, if upon the first attempt insufficient amount of blood was collected (<350 µL), the individual was excluded from the study. In total, two children (8%) were excluded because of insufficient pooled capillary blood sampling, and two more children (8%) were excluded for venous blood sampling due to parental refusal for this blood collection after the capillary blood sample was already extracted. No blood samples from adult female or elderly participants were excluded for these reasons.

Individuals were selected through a four-stage sampling design. In the first stage, basic geostatistical areas were identified using a proportional probability of the population under four years old. In the second stage, four street blocks were randomly selected; in the third stage, six houses in each block were selected by systematic sampling. Finally, in the fourth stage, one child of each year of age between one and four (12–23 months, 24–35 m, 36–47 m and 48–59 m), one adult female, and one older adult were selected in each household. 

Demographic information was collected using ad hoc questionnaires. All methods were performed in accordance with the approved protocol, relevant guidelines, and regulations of our review board of the National Institute of Public Health in Mexico.

### 2.2. Blood Collection Procedures and Laboratory Measurements

#### 2.2.1. Capillary Blood Samples

Participants were randomly allocated into one of two groups according to the procedures and techniques summarized in Figure 1. In the DCB group, procedures were as follows: a finger prick was made on the left hand using high-flux BD lancets (pink for children—Mexico catalog key: 080.574.0032, code number: 366593, and blue for adults -Mexico catalog key: 080.574.0032, code number: 366594), and the first drop of blood was wiped away with a sterile 2 × 2 gauze. Then, the second and third capillary blood drops were placed on two microcuvettes (batch: 1903391) to measure two readings of Hb in the HemoCue devices. In the PCB group, capillary blood samples were obtained as follows: a finger prick was made on the left hand with a lancet BD, as previously described. The first drop of capillary blood was wiped away with a sterile 2 × 2 gauze. The second drop and the subsequent drops were placed into the microtainer with K_2_-EDTA (code number: 365974) to make 350–500 µL. Once sufficient capillary blood was collected, the microtainer was rotated and inverted 20 times to assure the mixture of the capillary pool with the anticoagulant. A capillary tube (free mineral) was then inserted into the microtainers in a 45° angle to collect ≈40 µL of capillary blood and a drop of blood, from the capillary tube, was placed in a microcuvette to measure Hb in the HemoCue device. A second drop was placed in another microcuvette to have a duplicate determination. Then, the microtainer was capped, labelled, and stored at 5 °C to be sent to the central laboratory in Cuernavaca, Morelos, for Hb concentration measurement using the cyanmethemoglobin method in a hematology analyzer.

#### 2.2.2. Venous Blood Sample

All participants provided a sample of 4 mL of venous blood from the left arm, which was collected in vacutainer tubes with K_2_-EDTA as anticoagulant (BD, code number 368171). Then, the vacutainer was rotated and mixed 10 times to assure adequate mixture of the blood with the anticoagulant. A capillary tube was used to collect ≈40 µL of venous blood and two drops of blood were placed in two different microcuvettes (batch: 1903391) to measure in duplicate Hb in HemoCue 201+ devices. The remaining venous blood was capped, labelled, and stored at 5 °C to be sent to the central laboratory for Hb concentration measurement using the cyanmethemoglobin method, as well as using a clinical commercial hemocounter (ABX micros 60). 

Figure 1 shows the diagram of measurements undergone by the participants.

### 2.3. Laboratory Measurements

Samples of pooled capillary blood and venous blood were sent to the central laboratory the same day of sample collection. For both types of samples, Hb concentration was determined using the cyanmethemoglobin method [16], which we considered the gold standard, in the equipment Beckman Coulter Ac•T 5diff^®^ (Beckman Coulter Inc., Danaher Corporation, CA, USA) [17]. Hb concentration from venous samples was additionally quantified by a conventional hematology analyzer using a non-polluting generic lysing reagent (Diagon-Dya Lyse, code number: 20212) in the equipment ABX Micros 60^®^ (Horiba, Kyoto, Japan) according to the manufacturer’s manual [18]. Only one experienced technician handled the venous and pooled capillary blood samples and used the two hemocounters. 

### 2.4. Statistical Analysis

We eliminated only one result of PCB because it was clearly erroneous; the difference against the reference value was 9 g/dL; this may have been due to clotting. This may have been caused either for very slow collection of the blood or by insufficient mixing of the sample which favoring clotting.

The mean difference of the Bland and Altman plot between the same venous blood samples analyzed in HemoCue vs. cyanmethemoglobin was used as an estimate of the average systematic bias of the HemoCue devices. This mean value (0.31 g/d) was larger than the reported expected accuracy for this equipment by the manufacturer, and it was used as correction factor for all Hb measurements obtained from the HemoCues used in this study. The Pearson correlation coefficient was also used to determine the potential linear response of the methods used within the range of Hb concentrations measured in the studied population. The concordance correlation coefficient was used to evaluate the level of agreement between Hb measurements of the different capillary blood source groups against venous blood analyzed using the gold standard.

A 95% distribution interval for the differences between duplicated Hb HemoCue measurements from drop and pooled capillary, and venous blood samples was used for estimating the precision of the method. The 95% distribution interval was obtained using the mean difference ± 1.96 times the standard deviation (SD).

A linear mixed model was used to study mean differences in Hb measurements. In the fixed component of the model, mean differences were estimated between Hb measurements using HemoCue 201+ as compared with the gold standard; both sampling methods of capillary blood were included in comparisons and the model was adjusted by age and sex. In addition, the random component of the model allowed the estimation of variance components of the individual study subjects as natural variation, and the variability between the blood sample measurements in each subject as measurement error.

Statistical significance was set at alpha = 0.05 for a 95% confidence. All analyses were performed using *Stata Statistical Software* (Release 16. College Station, TX: StataCorp LLC, USA).

## 3. Results

Table 1 shows participant characteristics by capillary blood source group. No differences were observed by age or sex (groups of children and elderly). No differences were observed among the mean Hb concentration in venous blood analyzed using the cyanmethemoglobin method among the groups. Therefore, the DCP group was comparable to the PCB group.

### 3.1. Comparative Performance of Cyanmethemoglobin Method with the Hemocounter ABX Micros 60 Method

Figure 2 shows the correlation, the Bland Altman concordance plots between the cyanmethemoglobin method and the Hemocounter ABX Micros 60 using all venous blood samples. In the range from 8 to 16 g/dL Hb, the 95% confidence interval between results using these two methodologies was ±0.29 g/dL (95% CI −0.23, 0.38). Hemocounter ABX had a small positive and negligible average bias (0.07 g/dL) when compared to the cyanmethemoglobin method. Only one value was outside the confidence interval range, and it was below 1.0 g/dL. These results showed the high accuracy, precision, and reproducibility of these two methods, which are practically interchangeable. Moreover, these results demonstrate the reliability in the determination of Hb when estimated in good-performing equipment, and the variability introduced by different technicians is minimized. They also confirmed the validity of using the cyanmethemoglobin results with venous blood as the reference values.

### 3.2. Verification of the HemoCue vs. Cyanmethemoglobin Method Using Venous Blood (Systematic Bias Estimation)

To determine the systematic error of the HemoCue, we analyzed the results of each one of six HemoCue devices against the results with the same venous samples using the cyanmethemoglobin method. Appendix A shows the results, which revealed that the HemoCue devices that we used in our study had a mean positive bias of 0.31 g/dL when compared against cyanmethemoglobin results. Only one of our HemoCue apparatus (#6) showed a smaller average bias. These results suggest that our equipment was less accurate than the technical specification (±1.5%, i.e., ±0.10 to ±0.20 g/dL) claimed by the manufacturer. Nevertheless, the technical performance of the equipment is still good enough to be used as a point-of-care device and, once the average error is determined, this systematic bias can be used to correct the values produced by the HemoCue. We adjusted the results of the HemoCues by subtracting 0.31 g/dL.

### 3.3. Precision of Methods

Figure 3 summarizes the variation of differences between Hb duplicate measurements of the different blood source samples analyzed in HemoCues. Venous blood and pooled capillary blood produced results with similar variations. The 95% confidence interval for differences in the HemoCue using venous blood samples was ±0.5 g/dL (95% CI −0.56, 0.39), and ±0.6 g/dL (95% CI −0.60, 0.55) for PCB. However, the variation of consecutive drops of capillary blood from the same individual was much higher: ±1.2 g/dL (95% CI −1.30, 1.00), meaning that differences in paired blood drop samples were twice those for PCB as compared with the other two blood samples. Here, it is important to point out that some values of duplicates of DCP were as high as 2.0 g/dL (i.e., about 20% farther from the true Hb value). This difference is too large for having reliable estimations of Hb concentrations, especially when only one drop of blood is analyzed. 

### 3.4. Comparing Venous Blood and PCB by the Cyanmethemoglobin Metod

We estimated the mean Hb difference of PCB versus venous blood from the same individuals using the gold standard (i.e., the cyanmethemoglobin method). PCB produced a slightly higher Hb concentration value (of ≈0.1 g/dL) than venous blood. This difference appears to be biological. Capillary blood has a slightly higher concentration of Hb than venous blood. Appendix A summarizes the variation between capillary and venous blood samples analyzed with the reference method. The 95% CI interval of differences between PCB and venous blood using the reference method was ±0.8 g/dL, 95% CI (−0.74, 0.93), and which was larger than the variation of results with venous blood samples (±0.29 g/dL; 95% CI −0.23, 0.38) analyzed by two different hematology analyzers, and it is attributable to the intrinsic variation of the Hb concentration in the PCB or introduced during the collection and handling of the capillary blood samples.

### 3.5. Hb Measurement of Different Blood Sample Sources by HemoCue 

Figure 4 shows the Bland–Altman plots of Hb measurements using HemoCue in venous blood, PCB, and DCP against the measurements of the corresponding paired samples analyzed using venous blood in the cyanmethemoglobin method, before and after adjustment for the systematic HemoCue average bias. As expected, after bias adjustment, values of both venous blood and PCB in HemoCue showed negligible average biases (0.00 and −0.02 g/dL, respectively). However, despite adjustment, DCB maintained a higher positive average bias (0.425 g/dL). 

In the HemoCues, the variation of the Hb concentration results increased from venous blood to PCB and to DCP. Thus, the 95% confidence intervals were ±0.6 (95% CI; −0.65, 0.65), ±0.7, (95% CI; −0.75, 0.70), and ±1.6 (95% CI; −1.18, 2.02) g/dL, respectively. Several samples of DCP were 2.0 to 3.0 g/dL away from the reference value. This shows that samples using DCP risk producing highly variable results as high as 20–30%, which combined with the high difference of the average bias, makes these results unreliable. 

By age group, higher dispersions were observed in DCB in all age groups, being the highest in children, followed by older adults, and then adult females. (Appendix A). In all age groups, the variance ratio test in DCB-based Hb measurements were statistically different from the dispersion of differences in PCB-based Hb measurements (*p* < 0.05) as well as the dispersion in venous-based values (*p* < 0.001). The variance ratio test in PCB/venous blood was different in children (*p* = 0.045), but not in adult females (*p* = 0.4327) nor older adults (*p* = 0.999).

Table 2 summarizes the comparative data of all Hb measurements using HemoCue versus the cyanmethemoglobin method, before and after adjustment using the systematic bias of the HemoCues. PCB showed a similar Hb mean difference to venous blood, and better precision than the DCB group. Higher concordance was observed before and after bias adjustment for the PCB results (0.93 and 0.96) than DCB results (0.76, 0.82). In both groups, as expected, venous blood analyzed with HemoCue yielded the highest correlation and concordance and lowest Hb difference as compared with venous blood analyzed with the cyanmethemoglobin method. Results by age group are presented in Appendix A.

Appendix A shows the frequency distribution of differences around the mean, before and after adjustment, using venous blood, PCB, and DCB. Adjustment for the HemoCue systematic bias was required independent of blood sample source, but results for DCB continued being erroneous despite adjustment. 

### 3.6. Mixed Linear Regression Model for Differences in Hb Measurements

Results for the adjusted mixed linear regression model are presented in Appendix A. Mean difference of Hb from the PCB group did not vary significantly from venous blood HemoCue measurements (β = 0.02, *p* = 1.00). This means that, despite a slightly larger variation of the results than those with venous blood, PCB may be useful for determining Hb concentration and is therefore a type of sample that may be collected in population-based surveys when collecting venous blood is not possible. 

The two Hb measurements of DCB were equally different from the reference value, as shown in the Bland–Altman analysis (Appendix A). This is evidence that the DCB measurements have higher random variation and, hence, produce less reliable results. By considering the average of the two Hb measurements using DCB, dispersion of Hb did not improve the estimation (Appendix A).

A marginally significant difference was observed for both Hb in capillary blood using the cyanmethemoglobin method against Hb results of the paired venous blood using the ABX Micros 60 hemocounter, as compared with venous blood using cyanmethemoglobin (β = 0.09, *p* = 0.09 and β = 0.08, *p* = 0.059, respectively).

Random effects from the mixed model showed that natural variation (by subject) represented over 90% of total variance, and 82.5% in children. This implies that measurement error is higher in children (17.5%) than in other age groups analyzed (3% for adult females and 7.3% for older adults). This emphasizes the importance of proper sampling procedures and well-trained personnel for blood sample collection in children (Appendix A).

## 4. Discussion

Our results include several major findings: (a) PCB samples, in our hands, showed acceptable performance in Hb measurement using HemoCue 201+ when compared with venous blood samples analyzed in the same equipment; (b) DCB showed the largest inaccuracy and imprecise Hb values as compared with venous blood, when the latter is analyzed in HemoCue or with the reference method (cyanmethemoglobin); (c) a higher variation of Hb results was found in DCB than PCB and venous blood, which implies that variation is random and variation cannot be corrected, which in our study was as high as 2.0 g/dL; (d) the HemoCue 201+ device in our study showed an average positive bias (that is a systematic error) when the results of the same venous samples were compared against the reference method, which can be used to correct the results of the HemoCue; and finally (e) the HemoCue measurements system (considering the measurement error from personnel) showed a good performance. The low variation (7%) was attributed to Hemocue and personal skills in the Hb measurement, being the main source of variability, the individual heterogeneity of the blood samples, as expected. Variation from measurement error in the Hemocue measurement system was the highest in children (17.7%) compared with other groups, which reflects the challenge of obtaining good quality blood samples in this population group. 

As the results of Hb concentration values estimated using DCB in the HemoCue varied widely from the true values calculated using venous blood, they might cause errors in the diagnosis of anemia both in individuals and populations. This is evidence that DCB is not a suitable sample for measuring Hb concentration as the errors can be as high as 20–30% of the true value. This finding corroborates Bond’s report [12] that documented a coefficient variation 3.4 times for Hb for subsequent drops than in venous blood.

Bond et al. [12] suggested that Hb measurement using several drops of blood is necessary to reduce variation, but this is neither practical nor economically feasible. In our study, two measurements from capillary drops neither reduced the variation nor produced acceptable results. 

Our results support the conclusions by Conway et al. [11], who recommended that finger prick for collecting drops of capillary blood to estimate Hb values should not be used.

PCB, analyzed with the cyanmethemoglobin method, produced a slightly higher difference in Hb concentration of ≈0.1 g/dL than venous blood, but is small enough to not necessitate correction. This higher Hb concentration is associated with capillary blood instead of venous blood. The high concordance of Hb from PCB versus venous blood found in our study highlights that PCB may be considered an acceptable alternative for estimating Hb using point-of-care devices, in circumstances when the collection of venous blood is challenging. Dasi et al. [13] reported a similar finding: a concordance of 0.97 for Hb concentration from PCB measured by a portable autoanalyzer versus venous blood measured using a traditional clinical hematology analyzer. The better precision of Hb measurement from PCB data may be explained by the low variability due to collection of a greater amount of blood drops as compared with single drops of blood. Nevertheless, PCB requires good training and standardization of personnel who collect the blood sample (as discussed below). It includes control of the time duration in collecting the sample, and good use of postural and mixed procedures [19]. Six of our PCB samples produced values that were out of the expected range, which suggest problems with those samples such as coagulation due to possible failure to adequately mix the blood or a very slow collection of the blood drops. To narrow the ratio of EDTA to blood, our study collected 350–500 µL of blood.

One study performed in Guatemala and Honduras showed that the HemoCue 201+ had negative systematic bias, which was the opposite tendency to what we observed in our study [20]. Therefore, the systematic bias of the HemoCue equipment may depend on the apparatus and not on the model. Thus, the performance of each device should be verified before use and its performance examined periodically. Calibration to manufacturer standards is insufficient, as well as only checking the optical conditions of the apparatus [3]. It is also important to confirm the accuracy of measurement against blood samples. The latter is especially important if the apparatus is intended for use in population-based surveys where small deviation on the true values may have important implications for calculating the Hb concentration distribution in the population, and therefore the estimation of the anemia prevalence. Systematic bias of the HemoCues can be easily corrected, by prior adjustment of the results of the apparatus based on paired results of blood standards analyzed in the HemoCue and in a reference method. 

Some studies have documented higher average Hb values (in the range of 0.3 to 0.8 g/dL) using HemoCue 201+ against results of a hemocounter [21,22], while others have not found a bias in the equipment. Our data came from a field setting scenario, where participants were interviewed at home in conditions outside clinical settings, where conditions of humidity and temperature affect the stability of the HemoCue cuvettes and electronic performance of the device [23,24]. A previous study carried out in San Luis Potosí, Mexico using HemoCue 201+ reported failures in completing Hb readings in locations with high temperatures (>35 °C) and humidity (Berenice Gaona, personal communication, data not published). In this study, except for the slight systematic positive bias of the HemoCue 201+, the equipment performed well.

Different sources of error may affect the accuracy and precision of the Hb measurements [24]. Well-trained and standardized personnel are crucial and minimize potential error in Hb determination in field studies. In the present study, personnel were instructed to puncture only once (in the finger for capillary blood or the arm for venous blood) to avoid participant exclusion, especially due to the refusal of mothers to allow their children to participate. Our success rate in collecting PCB was very high (97.3%) in all samples in comparison with a study carried out in Honduras [20], where non-trained health or medical personnel were involved. In that study, PCB samples could not be collected in a high proportion of children (38/46, 83%) or some adult females (3/24 = 12%). We attribute the success rate of our study, both in PCB and venous blood samples in children and adults, to the time invested (around two weeks) in training, sensitization, and standardization of our personnel for collecting and handling blood samples in field settings. These are important skills that must be assured for the reliable human measurement of Hb, independently of the method that is selected. 

Other sources of error may occur at the time of loading blood into the HemoCue micro-cuvettes, which increases the magnitude of error using these devices [19]. This may be even worse if inexperienced personnel are involved.

The following should also be considered when interpreting our results and comparing them to conflicting data reported in population-based surveys: we used high-flow BD-type lancets, which are characterized by a higher number of bezels (pentapoint) and allow high blood flux. Most national surveys use low-flow lancets that cause a shallow wound (one bezel), likely with the aim for less invasive methods. However, the one bezel lancet may restrict the flux of red blood cells and therefore increase inaccuracy and variation in Hb concentrations estimated using DCB. Our study did not examine the effects of using low-flow lancets.

Here, it is important to note that our results differ from one previous validation study using the HemoCue (B-Hemoglobin model) carried out in a hospital in Mexico [25]. In that study, only the average bias among the studied groups was assessed, and therefore the results were considered as acceptable. As the Hb concentration is interpreted based on thresholds both for individuals and populations, the most important is to assess the variation among results and not the average of them in the studied groups. Nevertheless, larger variation from DCB, particularly in children, found in that study was consistent with our results [25]. Moreover, the previous study was performed in a clinical setting (hospital), while our study was implemented in a field setting where external factors may have affected the results.

In summary, it is important to emphasize that the main reason for errors in the determination of Hb concentrations using HemoCues is not the systematic bias of the apparatus but the large and random variation if using DCB. The assumption that DCB is suitable for estimating Hb concentration is incorrect. Our results confirm previous recommendations to discourage the use of single drops of capillary blood collected by finger prick for Hb measurement [11,12,19].

To our knowledge, this is the first study to evaluate the results of Hb concentration measurements using the HemoCue in a field setting scenario with different blood sample sources: venous blood, PCB, and DCB. Furthermore, it is common that the performance of HemoCues is not verified using venous blood and a reference methodology; our results demonstrate the importance of estimating the systematic bias of each HemoCue apparatus against venous blood that is further analyzed using a clinical hemocounter.

Our findings may partially explain the discrepancy in estimations of anemia prevalence among surveys when verification of the equipment is uncommon, training may be insufficient, and the blood samples are mostly collected by DCB. Our results may also explain why a recent review of Hb data to estimate the evolution of anemia prevalence in national surveys from 2000 to 2019 did not find any progress in its reduction [26]. Most of the information used in this study came from DCB. Therefore, if the accuracy of the equipment is not checked and the precision of the results is poor, changes along years and among population strata will be difficult to detect. The determination of anemia prevalence in both individuals and in populations requires the use of reliable procedures and methodologies as it has been suggested by Nieves Garcia-Casal et cols [27], because erroneous results may lead to flawed interpretations. 

## 5. Conclusions

Venous blood is the ideal type of blood sample to determine Hb concentration using HemoCue and similar point-of-care devices. However, when the collection of venous blood is not possible, PCB samples may be a valid alternative if it is collected by experienced personnel. Hb concentration estimated in samples using DCB should be discontinued for the determination of Hb concentration in both individuals and populations, since the random error associated with the collection of DCB is high even in experienced personnel.

Nevertheless, it is important to bear in mind that measuring Hb concentration is insufficient for understanding the anemia prevalence. It is also essential to identify the etiological factors of anemia, for which the collection of sufficient amounts of blood is needed to assess the associated biomarkers. In addition, each point-of-care apparatus must be checked periodically against reference methods as systematic bias may exacerbate the variation associated with the ability of the personnel that uses them.

## Figures and Tables

**Figure 1 nutrients-14-05346-f001:**
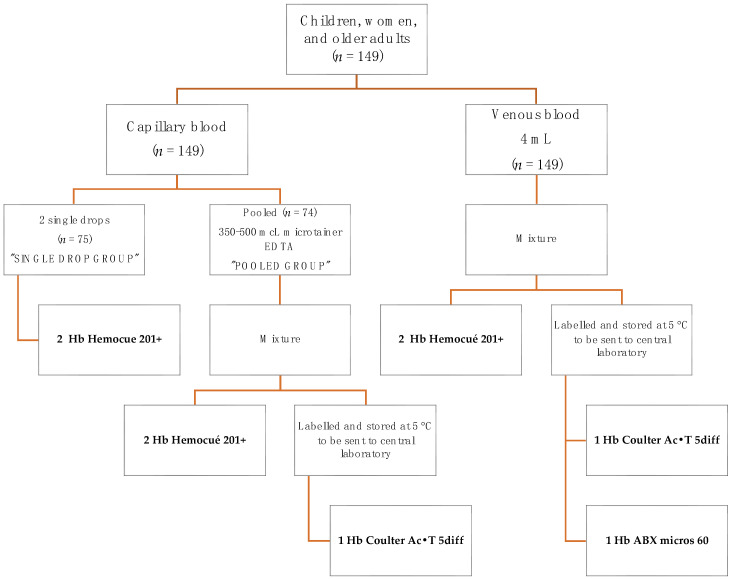
Procedures and measurements.

**Figure 2 nutrients-14-05346-f002:**
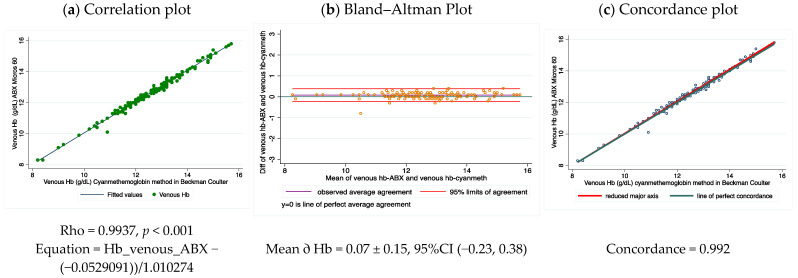
Correlation (**a**), Bland–Altman (**b**) and Concordance (**c**) plot for Hb concentrations by venous blood samples using the cyanmethemoglobin method and the HemoCounter ABX Micros 60.

**Figure 3 nutrients-14-05346-f003:**
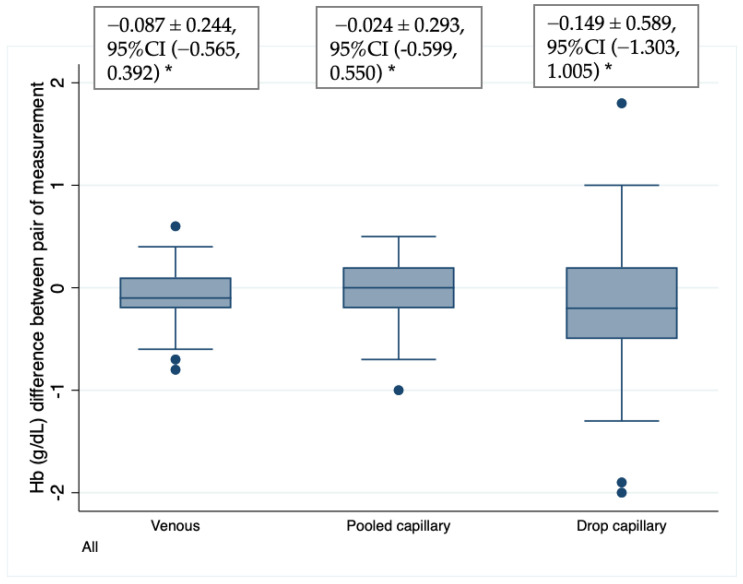
Box plots of Hb difference * (g/dL) between measurement duplicate analyzed using the HemoCue 201+. * Bland–Altman Mean ± SD; (95% CI).

**Figure 4 nutrients-14-05346-f004:**
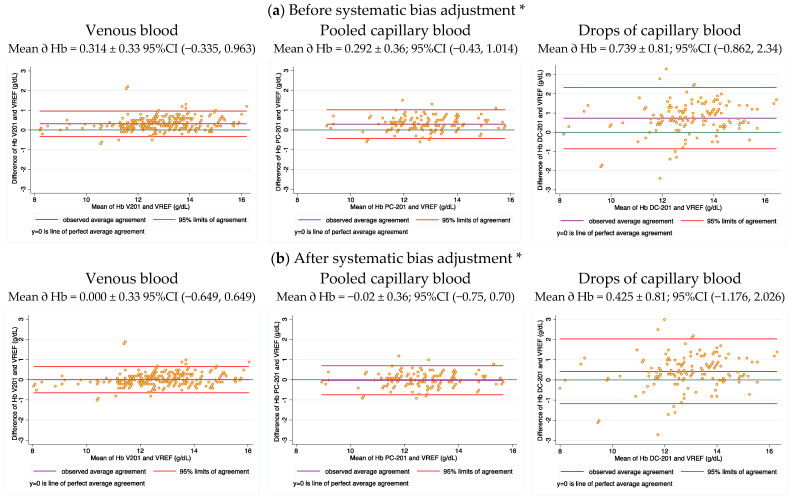
Bland–Altman plots of Hb concentration determined using the HemoCue in different blood sample sources versus using the cyanmethemoglobin method with venous blood. (**a**) Data before adjustment using the systematic bias *, (**b**) data after adjustment *. * The values of the HemoCue were adjusted by subtracting the average systematic bias of 0.314 g/dL estimated in the Bland–Altman plots that compared paired results of venous blood in the HemoCue and with the cyanmethemoglobin method. Abbreviations: V201 = Hb venous analyzed in Hemocue 201. VREF = Hb venous analyzed with the cyanmethemoglobin method. PC-201 = Pooled Capillary analyzed in Hemocue 201+. DC-201 = Drop Capillary analyzed in Hemocue 201+.

**Table 1 nutrients-14-05346-t001:** Descriptive characteristics of study participants.

Characteristic	Drop Capillary Blood (*n* = 75)	Pooled Capillary Blood (*n* = 74)	*p*-Value
Age (years, mean ± SD, (range))	35.5 ± 30.5 (1–90)	35.7 ± 28.7 (1–90)	0.974
Age group			
1 to 4 y	33.3	32.4	
18–45 y	33.3	33.8	
≥60 y	33.3	33.8	0.993
Sex (female, %)	70.6	64.4	0.449
Hb * (g/dL) (range)	12.7 ± 1.4 (8.2 to 15.6)	12.6 ± 1.3 (9.2 to 15.7)	0.768

* Measured in venous blood by cyanmethemoglobin.

**Table 2 nutrients-14-05346-t002:** Comparative results of differences in Hb concentrations by blood sample sources analyzed using the HemoCue 201+ versus venous blood samples analyzed using the cyanmethemoglobin method.

Group		Blood Sampling Method in Hemocue	Concordance	Pearson Correlation	Relative Bias	Hb Mean Difference (g/dL)
“Pooled” Capillary Blood	Before bias adjustment	Venous	0.95	0.97	0.97	0.26
		Pool (first and second lecture)	0.93	0.96	0.97	0.29
		Pool (first lecture)	0.94	0.96	0.97	0.28
		Pool (second lecture)	0.93	0.95	0.97	0.3
	After bias adjustment	Venous	0.98	0.98	1.00	−0.05
		Pool (first and second lecture)	0.96	0.96	1.00	−0.02
		Pool (first lecture)	0.97	0.97	1.00	−0.03
		Pool (second lecture)	0.95	0.95	1.00	−0.1
“Drops” of Capillary Blood	Before bias adjustment	Venous	0.94	0.97	0.97	0.36
		Drop (second and third drop)	0.77	0.86	0.89	0.74
		Drop (second drop)	0.79	0.87	0.91	0.66
		Drop (third drop)	0.75	0.86	0.87	0.81
	After bias adjustment	Venous	0.97	0.97	0.99	0.05
		Drop (second and third drop)	0.83	0.86	0.96	0.43
		Drop (second drop)	0.84	0.87	0.97	0.35
		Drop (third drop)	0.81	0.86	0.94	0.50

## Data Availability

The datasets used and/or analyzed throughout the current study are available from the corresponding author, T.S.-L., upon reasonable request.

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
