# Peer review of "Drops of Capillary Blood Are Not Appropriate for Hemoglobin Measurement with Point-of-Care Devices: A Comparative Study Using Drop Capillary, Pooled Capillary, and Venous Blood Samples"

_nutrients, 2022, doi:10.3390/nu14245346_

Round 1

Reviewer 1 Report

Dear Authors

You present an interesting research with very useful results. The only recommendation I have is that I think that there are self-citations that are not appropriate in the context.

The study is well designed and written. I could not find any gaps in the analysis. It is well known that the Hb measurement by "finger drop" is not as accurate as venous samples, however, in special occasions it is the best approach and studies like the one presented are lacking in literature. The sample is small but children are also included.

Author Response

Dear Editor and Reviewers,

We have addressed the comments from the reviewers. The changes are highlighted in the document, which it was reviewed by a native English speaking. Here are the point by point response to the Reviewer`s comments. We appreciate the time and effort you have spent to share your insightful comments that substantially improved our manuscript.

We look forward to your kind consideration,

On behalf of all co-authors

Teresa Shamah-Levy
Corresponding author

Reviewer 1

You present an interesting research with very useful results. The only recommendation I have is that I think that there are self-citations that are not appropriate in the context.

R= Thank you for the comment. We reviewed the document again, and we removed three self-citations from the Introduction and discussion section.

The study is well designed and written. I could not find any gaps in the analysis. It is well known that the Hb measurement by "finger drop" is not as accurate as venous samples, however, in special occasions it is the best approach and studies like the one presented are lacking in literature. The sample is small but children are also included.

R=Thank you for the comment.

Reviewer 2 Report

Results and dicusssion/conclusion must be clearly increased and better explained! Good paper but I would like it to be rewritten in a clearer way! Please submit a clearer version of the paper!

Author Response

Dear Editor and reviewer,

We have addressed the comments from the reviewers. The changes are highlighted in the document, which it was reviewed by a native English speaking. Here are the point by point response to the Reviewer`s comments. We appreciate the time and effort you have spent to share your insightful comments that substantially improved our manuscript.

We look forward to your kind consideration,

On behalf of all co-authors

Teresa Shamah-Levy
Corresponding author

Reviewer  2

Results and dicusssion/conclusion must be clearly increased and better explained! Good paper but I would like it to be rewritten in a clearer way! Please submit a clearer version of the paper!

R=Thank you for the comment. We agree that the paper is written using many technical terms but it expresses the main concern of the measurement errors of Hb in capillary drops. We reviewed it again and we made some adjustments to the interpretation in order to clarify the discussion of the findings. In the results section, the changes were minor since results were described as showed in the analysis.
